# Discovery and Characterization of a Dual-Function Peptide Derived from Bitter Gourd Seed Protein Using Two Orthogonal Bioassay-Guided Fractionations Coupled with In Silico Analysis

**DOI:** 10.3390/ph16111629

**Published:** 2023-11-20

**Authors:** Wei-Ting Hung, Christoper Caesar Yudho Sutopo, Mei-Li Wu, Jue-Liang Hsu

**Affiliations:** 1Department of Food Science, National Pingtung University of Science and Technology, Pingtung 912, Taiwan; milahung4123@gmail.com (W.-T.H.); mlwu@mail.npust.edu.tw (M.-L.W.); 2Department of Tropical Agriculture and International Cooperation, National Pingtung University of Science and Technology, Pingtung 912, Taiwan; christopercaesar@gmail.com; 3Department of Biological Science and Technology, National Pingtung University of Science and Technology, Pingtung 912, Taiwan

**Keywords:** bitter gourd seeds, ACE inhibitory peptide, DPP4 inhibitory peptide, in silico analysis, LC-MS/MS

## Abstract

The hydrolysate of bitter gourd seed protein, digested by the combined gastrointestinal proteases (BGSP-GPs), exhibited the most potent inhibition on angiotensin-I-converting enzyme (ACE) with an IC_50_ value of 48.1 ± 2.0 µg/mL. Using two independent bioassay-guided fractionations, fraction F5 from reversed-phase chromatography and fraction S1 from strong cation exchange chromatography exhibited the highest ACE inhibitory (ACEI) activity. Three identical peptides were simultaneously detected from both fractions and, based on the in silico appraisal, APLVSW (AW6) was predicted as a promising ACEI peptide. Their dipeptidyl peptidase-IV (DPP4) inhibitory (DPP4I) activity was also explored. The IC_50_ values of AW6 against ACE and DPP4 were calculated to be 9.6 ± 0.3 and 145.4 ± 4.4 µM, respectively. The inhibitory kinetics and intermolecular interaction studies suggested that AW6 is an ACE competitive inhibitor and a DPP4 non-competitive inhibitor. The quantities of AW6 in BGSP-GP hydrolysate, fractions F5 and S1, were also analyzed using liquid chromatography–tandem mass spectrometry. Notably, AW6 could resist hydrolysis in the human gastrointestinal tract according to the result of the simulated gastrointestinal digestion. To the best of our knowledge, this is the first discovery and characterization of a dual-function (ACEI and DPP4I activities) peptide derived from bitter gourd seed protein.

## 1. Introduction

In a mammal’s blood pressure regulation system, the angiotensin-I-converting enzyme (ACE) cleaves angiotensin I (Ang-I) into the vasoconstrictor angiotensin II (Ang-II), reducing vasodilation and diuretic effects by altering bradykinin into an inactive form. Simultaneously, ACE2, a homologue of ACE, counteracts the vasoconstrictor effects by converting Ang-II into Ang 1–7 (a vasodilator). ACE2 also transforms Ang-I into Ang 1–9, and ACE hydrolyzes Ang 1–9 into Ang 1–7, resulting in a vasodilation effect [1]. However, Ang 1–7 has a short half-life of only half an hour in the human body, and its concentration peaks in plasma one hour after intracutaneous injection [2]. Hitherto, the inhibition of ACE has been a pharmaceutical approach for hypertension management. Motivated by the discovery of the first ACE inhibitory (ACEI) drug derived from a Brazilian snake venom peptide (captopril) [3], and the development of commercialized milk drinks containing antihypertensive peptides (such as Ameal S™ by Calpis Co., Ltd., Tokyo, Japan and Evolus™ by Valio Ltd., Helsinki, Finland) [4], there has been a growing interest in exploring ACEI peptides derived from natural protein sources. For instance, ACEI peptides derived from natural protein sources such as scorpion venom [5], calf thymus [6], *Saccharomyces cerevisiae* [7], animal products (meat, fish, blood, milk, dairy products, and egg) [8], as well as plants (seeds, pulses, mushrooms, and vegetables) [9] have been reported recently.

Hypertension patients show a 2–3 times higher risk of developing diabetes compared to patients without hypertension [10]. Additionally, the number of youths (those under 20 years old) in the United States with type 2 diabetes is projected to increase by 69% from 2017 to 2060 [11]. To date, various therapeutics for type 2 diabetes have been proposed, including the inhibition of dipeptidyl peptidase-IV (DPP4). DPP4 rapidly converts glucagon-like peptide-1 (GLP-1) and glucose-dependent insulinotropic peptide (GIP) into an inactive form, causing a deficiency in insulin secretion commonly found in type 2 diabetes patients. Inhibiting DPP4 may extend the half-life of incretins and prolong their insulinotropic effect [12]. Researchers have emphasized the potential of peptides derived from food proteins as natural DPP4 inhibitors that may have complementary or additive effects in managing type 2 diabetes. For instance, some natural DPP4 inhibitory (DPP4I) peptides derived from plant or animal proteins, such as dark tea [13], largemouth bass [14], and various vegetables (soy, pea, potato, quinoa, chickpea, lentil, and lupine) [15], have been reported. Diabetes can lead to complications in hypertensive patients, and hypertension can lead to complications in diabetes patients [16]. Therefore, a natural peptide with both ACE and DPP4 inhibitory activity might be beneficial in slowing down the impact of these complications.

Bitter gourd is a common vegetable and herb plant in Asia, particularly Taiwan. It has also been recorded as a therapeutic herb that dates back over 2000 years in Chinese materia medica, the Shennong Ben Cao Jing. Bitter gourd (*Momordica charantia*) seeds are a by-product of the bitter gourd food-processing industry, yet they contain relatively high amounts of protein [17]. Previous studies on bitter melon have revealed a wide range of biological activities, e.g., antioxidant, antimycotic, antihyperglycemic, antiobesity, stomachic, anticancer, hypotensive, and anticholesterol activities [18]. Bitter gourd fruit pulp extract has demonstrated hypoglycemic activity in type 2 diabetic patients [19]. In our previous study, bitter gourd seed protein hydrolysate and its derived peptide exhibited ACEI activity and significantly lowered the systolic blood pressure of spontaneously hypertensive rates [20]. However, there has been limited exploration of ACEI peptides with DPP4I activity derived from bitter gourd seed protein derivatives.

In this study, ACEI peptides from bitter gourd hydrolysate were screened independently using two liquid chromatography phases: reversed-phase high-performance liquid chromatography (RP-HPLC) and strong cation exchange (SCX) chromatography. The peptides from the most active fractions in both RP-HPLC and SCX separations were identified through high-resolution mass spectrometry (HR-MS/MS) analysis. Identical peptides that were concurrently found in the most active fractions of RP-HPLC and SCX separations were considered ACEI peptide candidates. BIOPEP and PeptideRanker databases were used to appraise the biological activity of ACEI peptide candidates, while the ToxinPred database was employed to predict their toxicity. The ACEI peptide candidate with the highest biological activity score, determined by both BIOPEP and PeptideRanker, was synthesized for further biological activity confirmation and characterization. The IC_50_ and its inhibition mechanism toward ACE were confirmed. The IC_50_ against DPP4 and its inhibition mechanism were also determined to explore the potency of this peptide. The molecular-level interaction of an active peptide with ACE and DPP4 was rationalized via molecular docking simulation. Furthermore, the resistance of the peptide toward the simulation of gastrointestinal protease digestion was evaluated, and the quantity of active peptide in hydrolysate and the fractions (RP-HPLC and SCX) were calculated using the multiple reaction monitoring (MRM) mode in a liquid chromatography–tandem mass spectrometry (LC-MS/MS) analysis. The discovery of an ACEI peptide with DPP4I activity derived from the bitter gourd seed protein holds promise for enhancing hypertension management in patients with type 2 diabetes. Additionally, this discovery has the potential to increase the value of bitter gourd seeds as a natural source of bioactive peptides. To the best of our knowledge, this study represents the first exploration of an ACEI peptide with DPP4I activity derived from bitter gourd seed protein.

## 2. Results and Discussion

### 2.1. ACE Inhibitory Assay of Bitter Gourd Seed Protein (BGSP) Hydrolysate

The ACE inhibitory (ACEI) activities of bitter gourd seed protein (BGSP) hydrolysates under 3 kDa were investigated. Although all the BGSP hydrolysates showed potent ACEI inhibitory activity (Figure 1A), the hydrolysate from gastrointestinal protease (GP) digestion exhibited superior ACEI activity compared to others (pepsin, trypsin, and α-chymotrypsin). Previous studies implied that several aspects, such as the protein substrate, specificity of protease, hydrolysis conditions, and hydrolysis degree, influenced the biological activities of hydrolysates [21]. The BGSP hydrolysate was monitored using UV–Vis RP-HPLC (λ = 214 nm) at a concentration of 50 µg/µL, and the HPLC chromatogram of the BGSP hydrolysate generated by GP (BGSP-GP) showed more intense peaks of short-chain peptides (under 3 kDa) than others (Appendix A). Likewise, the BGSP-GP hydrolysate exhibited the highest hydrolysis degree (~20%) compared to the other BGSP hydrolysates (Appendix A). This result indicates that gastrointestinal protease (GP) digestion could generate more short-chain peptides from BGSP and significantly influence their biological activity more than other proteases. Using several concentrations (0.02–033 µg/µL) of BGSP-GP hydrolysate on a logarithmic scale, the half-maximal ACEI concentration (ACEI IC_50_) was calculated by non-linear regression to give a value of 48.1 ± 2.0 µg/mL (Figure 1B). Compared with the ACEI IC_50_ value of *Momordica cochinchinensis* seed protein GP hydrolysate (70 ± 4 µg/mL) reported previously [22], the BGSP-GP hydrolysate exhibited stronger ACEI activity.

### 2.2. Bioassay-Guided Fractionation of BGSP-GP Hydrolysate

Bioassay-guided fractionation is crucial for effectively screening the active peptide from the peptide mixtures. Reversed-phase (RP) chromatography is extensively used for separating peptides from protein hydrolysates [23], and a C_18_ column is preferable for separating peptides (under 5 kDa) resulting from the protease digestion of proteins [24]. The BGSP hydrolysate (under 3 kDa) digested by gastrointestinal protease (GP) was separated using a UV–Vis HPLC equipped with a C_18_ column. As shown in Figure 2A, the fractions were collected every five minutes into 12 fractions (F1–F12), and the fraction F5 from RP-HPLC fractionation exhibited the most potent ACEI activity (77 ± 1%) among other fractions. The active peptide with ACEI activity could not be easily recognized in the BGSP-GP hydrolysate fraction F5. This is because several peptides were observed after identifying peptides using LC-MS/MS analysis. Strong cation exchange chromatography (SCX) has been widely used to fractionate proteins or peptides. Hence, to increase the certainty of recognizing the ACEI peptide, bioassay-guided SCX liquid chromatography was conducted individually to fractionate the peptide mixture from the BGSP-GP hydrolysate. The BGSP-GP hydrolysate S1 fraction from SCX fractionation showed significantly potent ACEI activity (57 ± 2%) compared to other SCX fractions (Figure 2B).

### 2.3. Peptide Identification and In Silico Analysis of ACEI Peptide Candidates

The peptide mixtures in the most active fraction from RP-HPLC (fraction F5) and SCX (fraction S1) fractionations were analyzed using liquid chromatography–tandem high-resolution mass spectrometry (LC-HR-MS/MS). Then, automated de novo peptide sequencing and database matching were performed in PEAK X+ version 10.5 software to interpret the MS and MS/MS peptide spectra for peptide identification. Forty-four peptide sequences from RP-HPLC fraction F5 and eighteen from SCX fraction S1 were identified, as shown in Appendix A, respectively. Three identical peptides were observed from both the F5 and S1 fractions: APLVSW (AW6), EPTTSDVVVAGEFDQGSGSMR (ER21), and ATISLENSW (AW9). Computational analyses using the BIOPEP and PeptideRanker databases were performed to predict the biological activity of these ACEI peptide candidates. The toxicity of these peptides was also predicted using ToxinPred, and all the peptides were appraised as non-toxic (Table 1). APLVSW (AW6) scored highest in the BIOPEP and PeptideRanker biological activity appraisals. Also, AW6 is simultaneously recognized in the F5 fraction from RP-HPLC and the S1 fraction from SCX, making it conceivably the major peptide credited for potent ACEI activity. To confirm the ACEI activity and its identity, AW6 was chemically synthesized. The synthetic AW6’s purity of 99% was verified using RP-HPLC (Appendix A). Additionally, Figure 3 displays very similar retention times of synthetic AW6 (t_R_ 16.62 min), SCX fraction S1 (t_R_ 16.63 min), and RP-HPLC fraction F5 (t_R_ 16.64 min) with identical MS ion spectra (m/z 672.4), implying additional confirmation of this peptide candidate.

### 2.4. AW6 Half-Maximal ACE and DPP4 Inhibitory Concentration

The 50% inhibitory concentration (IC_50_) is the most common and empirical indicator to measure the efficacy of a substance [25,26]. A non-linear interpolation was generated by seven logarithmic concentrations of synthetic AW6 versus their ACEI activities to determine the 50% ACE inhibitory concentration of AW6 (ACEI IC_50_). The ACEI IC_50_ of AW6 was determined to be 9.6 ± 0.3 µM, as shown in Figure 4. The ACEI IC_50_ of AW6 seems comparable to other ACEI peptides derived from BGSP thermolysin hydrolysate (VY7, IC_50_ = 8.6 µM, VG8, IC_50_ = 13.3 µM) [20] and showed favorable inhibition compared to sicklepod seeds (FK6, IC_50_ = 16.8 µM) [27], barley (FF6, GF8, NF6, AM6, IC_50_ value range from 28.2–200 µM) [28], and rice bran (YK3, IC_50_ = 76 µM) [29], as previously reported. The half-maximal DPP4 inhibitory concentration (DPP4I IC_50_) of AW6 was also determined by a non-linear interpolation. Using seven logarithmic concentrations of AW6 versus their DPP4I activities, the AW6 DPP4I IC_50_ value is estimated to be 145.4 ± 4.4 µM, as shown in Figure 5. Referring to the DPP4 inhibitory IC_50_ value derived from other plant seeds, such as napin of rapeseed (IS5, EL8, PF5, KP6, IC_50_ value range from 52.2–162.7 µM) [30], soybean glycinin (IA7, IC_50_ = 106 µM), and lupin seed β-conglutin (LD9, IC_50_ = 228 µM) [31], AW6 derived from BGSP-GP hydrolysate showed comparable DPP4 inhibitory activity. It is suggested that peptides with N-terminal Ala, Gly, Ile, Pro, Met, Glu, and Val, and the presence of Pro or Ala at the second position of the N-terminal, are preferred for inhibiting DPP4 [32]. Moreover, peptides containing aliphatic and hydrophobic amino acids (Ala, Val, Leu, Ile, Met, and Pro) with C-terminal aromatic side chain peptides (Tyr, Phe, and Trp) showed a strong influence on inhibiting ACE [4]. AW6 (APLVSW) has aliphatic residues at its N-terminus, a proline at the second position, several hydrophobic residues in the middle, and an aromatic tryptophan at the C-terminus. These specific features might facilitate significant contributions to ACE and DPP4 inhibitory activity.

### 2.5. Inhibitory Mechanism of AW6 toward ACE and DPP4

The double-reciprocal plot (the Lineweaver–Burk plot) is constructed to elucidate the inhibitory mechanism of AW6 toward ACE and DPP4 based on the Michaelis–Menten constant (Km) and maximum velocity (Vmax) as the *X*- and *Y*-axis intercept of the primary plot, respectively. As shown in Figure 6, the Lineweaver–Burk plot suggested that AW6 inhibits ACE as a competitive inhibitor because of the stable Vmax and increased Km value. The unaffected Vmax values and increased Km in the absence or presence of AW6 indicated that AW6 could compete with the analogue substrate for binding to the ACE active site. Meanwhile, the inhibitory mechanism study of AW6 toward DPP4 through the Lineweaver–Burk plot indicated that AW6 was acting as a non-competitive inhibitor on DPP4 (Figure 7). The Km value is unchanged by an increase in AW6 concentration. However, the Vmax value decreases, which may have illustrated that the inhibitor had a lack of impact on the affinities of the DPP4 substrate. Therefore, this peptide could be predicted as a competitive ACE inhibitor by forming interactions with the ACE active site and a non-competitive DPP4 inhibitor by interacting with a vital group other than the active sites of DPP4.

### 2.6. Intermolecular Interaction Study of AW6 toward ACE and DPP4 Using Molecular Docking Simulation

The intermolecular interactions between the inhibitory peptide (AW6) and target enzymes (ACE and DPP4) were interpreted using the molecular docking simulation. The rationalization of intermolecular interaction between AW6 and human tACE (PDB code: 1O86) resulted in the lowest binding CDOCKER energy of −99.65 kJ/mol, as shown in Figure 8. The active catalytic site of ACE comprises the S1 pocket (Tyr523, Glu384, and Ala354), the S1′ pocket (Glu162), the S2′ pocket (Tyr520, Lys511, His353, and Gln281), and a Zn^2+^ as the cofactor coordinated with His383, His387, and Glu411 [33]. As summarized in Appendix A, AW6 formed four hydrogen bonds, one ionic interaction, and one π–π interaction with the residues of the ACE catalytic site. Also, AW6 formed interactions with two ACE non-catalytic sites, which may have given it more force to interact with ACE: Glu142 (two hydrogen bonds and one ionic interaction) and Asn70 (one hydrogen bond). Compared with the ACE–lisinopril binding, which is also considered an ACE active site, AW6 did not form interactions with two residues from the S1 pocket (Ala354 and Glu384) and two residues from the S1′ pocket (Glu162 and Gln281). Still, these interactions are consistent with the results of the inhibition mechanism study regarding the interaction of AW6 with ACE active site residues, indicating that AW6 inhibits ACE through a competitive mechanism. Meanwhile, the molecular docking simulation of AW6 and human DPP4 (PDB code: 1WCY) resulted in the lowest CDOCKER energy of −63.81 kJ/mol (Figure 9). Diprotin A (a competitive inhibitor and low-turnover substrate of DPP4) interacts with the S1 pocket (Tyr547 and Tyr631), the S2 pocket (Glu205, Glu206, and Tyr662), and the S1′ pocket (Arg125) of the DPP4 catalytic pocket [34,35]. Compared to the interaction of diprotin A and DPP4 (as shown in Appendix A), AW6 interacted with DPP4 residues other than the catalytic sites (Ser59, Ile407, Arg471, and Glu408), consistent with the inhibition mechanism study that showed that AW6 is a non-competitive DPP4 inhibitor. Hence, the molecular docking study of AW6 with ACE and DPP4 presented additional affirmation that AW6 is a competitive inhibitor of ACE and a non-competitive DPP4 inhibitor. 

### 2.7. In Vitro Simulated Gastrointestinal (SGI) Digestion Stability of AW6

In vitro ACEI activity of peptides is sometimes inconsistent with their in vivo antihypertensive activity, which is based on their stability toward gastrointestinal protease enzymes and brush-border membrane peptidases in the digestive system [36], and biologically active peptide products are usually administered orally. Thus, investigating the alterations in peptide activity caused by gastrointestinal digestion is crucial. AW6 was derived using gastrointestinal protease digestion. Theoretically, the same proteolytic enzyme will not cleave it. To emulate AW6 stability during human gastrointestinal digestion, AW6 was mixed with pepsin for 1.5 h at 37 °C (pH 2.0), continued by α-chymotrypsin and trypsin (pH 8.0) digestion for 3 h. The stability of AW6 was observed using LC-MS (Figure 10). As expected, the AW6 was not hydrolyzed during the simulation of gastrointestinal digestion. The leucine in the antepenultimate N-terminal of AW6 was not cleft by gastrointestinal protease (pepsin or α-chymotrypsin), which may be due to the presence of the N-terminal penultimate proline residue that could improve the SGI digestion resistance of AW6 [37]. Also, AW6 features a low molecular weight (<3 kDa) and usually shows more resistance during SGI digestion than peptides with a higher molecular weight [38]. Therefore, the resistance of AW6 during SGI digestion suggests that AW6 is a promising biologically active peptide candidate for developing functional foods and therapeutic agents with ACE and DPP4 inhibitory activity.

### 2.8. Quantification of AW6 in BGSP-GP Hydrolysate, Fraction F5 (RP-HPLC), and Fraction S1 (SCX)

The quantity of AW6 in the BGSP-GP hydrolysate, fraction F5, and fraction S1 are further important factors for industrial applicability and production quality control. AW6 was quantified using multiple reaction monitoring (MRM) analysis in a triple-stage quadrupole (TSQ) mass analyzer. The standard solution of synthetic AW6 (10 ng/µL) was infused into the TSQ mass analyzer in MRM mode to give an optimum precursor–product ion transition m/z 672.4→282.2 under 34 volts of collision-induced dissociation energy. A linear regression model (y = 0.7774x + 48207; R2 = 0.9993) was obtained from the standard calibration curve of AW6 (0.5 pM–20 µM), as shown in Appendix A. The amounts of AW6 per milligram in the BGSP-GP hydrolysate, fraction F5, and fraction S1 were calculated at about 1.8 ± 0.2, 9.8 ± 0.3, and 2.5 ± 0.2 µg, respectively (Table 2). The amount of AW6 in either fraction F5 or S1 is significantly increased compared to the BGSP-GP hydrolysate. Although the fraction F5 from RP-HPLC fractionation showed higher AW6 content than the fraction S1 from SCX fractionation, SCX fractionation still showed relatively fair purification efficiency to purify AW6 at about 1.4-fold compared to the hydrolysate. In comparison to other ACEI peptide content in the hydrolysate, 1 mg of BGSP-GP hydrolysate contains 1.8 µg APLVSW (AW6), which is higher than FPHAPWK (147 ng) in Cassia obstusifolia seed hydrolysate [27] and HLPLPLL (19.9 pg) in milk hydrolysate [39], but slightly lower than DHSTAVW (3 µg) in garlic hydrolysate [40]. Likewise, compared to other DPP4I peptides, APLVSW (AW6) content in 1 mg of hydrolysate is somewhat higher than IP (1.2 µg) and LP (1.7 µg) in rice bran hydrolysate [41].

## 3. Materials and Methods

### 3.1. Materials

Bitter gourd seeds were collected in Pingtung County, Taiwan. Trichloroacetic acid, boric acid, sodium lauryl sulfate, sodium hydroxide, sodium chloride, hydrogen chloride, dimethyl ketone, dimethylformamide, ethyl ether, acetonitrile, formic acid, and methyl alcohol were purchased from J. T. Baker (Phillipsburg, NJ, USA). Pepsin from hog stomach, trypsin from bovine pancreas, α-chymotrypsin from bovine pancreas, angiotensin-I-converting enzyme (ACE) from rabbit lungs, recombinant human dipeptidyl peptidase-IV (DPP4), hippuryl-histidine-leucine, GP-pNA, captopril, linagliptin, trifluoroacetic acid, diisopropylethylamine, ferulic acid, hippuric acid, and p-nitroanilide were obtained from Sigma-Aldrich Co. (Saint Louis, MO, USA). Wang resin and HBTU were acquired from Creo Salus (Louisville, KY, USA). Fmoc-amino acids and OxymaPure were purchased from CEM Co. (Matthew, NC, USA). A PURELAB^®^ water purifier (Lane End, High Wycombe, UK) was utilized to generate deionized water, and all other chemicals used in this study were analytical or HPLC grade.

### 3.2. Bitter Gourd Seed Protein (BGSP) Extraction

Briefly, 50 g of bitter gourd seed (BGS) was ground and mixed with 500 mL of 1% SDS solution. A digital ultrasonic homogenizer from Branson Ultrasonic (Terra Universal Inc., Fullerton, CA, USA) was set with a 30% duty cycle for 10 min with a pulse of 10 s on and off to disrupt the cell membranes of BGS. The supernatant was collected by 4000 rpm centrifugation for 15 min. Then, 20% trichloroacetic acid in cold acetone was added at a 1:1 (*v*/*v*) ratio, and the mixture was incubated for at least 12 h at a mean temperature of −20 °C. The precipitated protein was lyophilized (~21.9 g) and stored in a dehumidifying cabinet at room temperature.

### 3.3. Bitter Gourd Seed Protein Hydrolysate Preparation

Bitter gourd seed protein (BGSP) was separately hydrolyzed using α-chymotrypsin, pepsin, trypsin, and gastrointestinal protease. α-chymotrypsin, pepsin, and trypsin were individually added to BGSP with an enzyme-to-protein ratio of 1:50 (*w*/*w*). The hydrolysis conditions for trypsin and α-chymotrypsin were maintained at pH 8.0 using 25 mM NH_4_HCO_3_, while pepsin was conditioned at pH 2.0 using 35 mM sodium chloride buffer at 37 °C for 16 h. The gastrointestinal protease (GP) digestion was initialized using pepsin digestion with an enzyme-to-protein ratio of 1:50 (*w*/*w*) in 35 mM sodium chloride solution at pH 2.0, 37 °C for 16 h. Before α-chymotrypsin and trypsin were added with an enzyme-to-protein ratio of 1:100 (*w*/*w*), the pH was adjusted to 8.0 using 10 N NaOH. The hydrolysis was continued for 16 h at 37 °C. The enzymatic hydrolysis, after 32 h of reaction, was terminated by boiling for 10 min. The hydrolysates were centrifuged at 12,000 rpm for 15 min at 4 °C. The supernatant was filtered through a 3 kDa MWCO ultrafiltration membrane, and the salt residues from the hydrolysis reaction buffer were removed using an HYPERSEP retain PEP C_18_ cartridge (Thermo Fisher Scientific, San Jose, CA, USA). The desalted BGSP hydrolysates (under 3 kDa) were lyophilized and stored at room temperature in a dehumidifying cabinet.

### 3.4. Hydrolysis Degree Analysis of BGSP Hydrolysate

The hydrolysis degree (HD) of BGSP hydrolysates was determined using the 2,4,6-trinitrobenzensulfonic acid (TNBS) assay and carried out in a 96-well microplate as described by Sutopo C. C. Y. et al. [42], with a few adjustments. BGSP hydrolysate (5 µg/µL) was premixed with 1% SDS solution in a 1:10 ratio (v/v). Then, the mixture (15 µL) was transferred to a 96-well microplate. Then, 45 µL of 212.5 µM sodium phosphate (pH 8.2) and 45 µL of TNBS (0.05%) were added immediately. The reaction was maintained at 50 °C with shaking at 70 rpm in a dark environment for an hour. Then, 90 µL of 0.1 M hydrochloride acid was added to terminate the reaction. The absorbance was monitored using a PowerWave XS (Biotek^®^, Winooski, VT, USA) 96-well microplate reader at a wavelength of 340 nm. A calibration curve of L-leucine (0.2–2 mM) was generated to quantify the primary amines of amino acids liberated at a certain digestion time. The *HD* was calculated as follows:(1)$$HD=\frac{h_{t}}{h_{tot}}\times 100\%,$$

The *h_t_* is the amount of liberated amino acids after a certain digestion time, and *h_tot_* is the total amino acid amount of BGSP hydrolysates. The *h_tot_* was calculated to be 45.5 mM by the following: *h_tot_* = BGSP hydrolysate (5 µg/µL)/amino acid average molecular weight (110 g/mol).

### 3.5. Determination of Angiotensin-I-Converting Enzyme Inhibitory (ACEI) Activity and Its Half-Maximal Inhibitory Concentration

The ACEI activity was studied according to Cushman and Cheung [43], with a few modifications. First, 10 µL of the testing sample was premixed with 30 µL of hippuryl–histidine–leucine (HHL) and incubated at 37 °C for 5 min. Then, 20 µL of ACE (0.05 mU/µL) was added. Afterward, the reaction was held statically at 37 °C for 30 min and continued with shaking at 200 rpm for another 30 min. Then, 60 µL of 1 N hydrogen chloride was used to quench the reaction, and 10 µL of 0.25 µg/µL ferulic acid was mixed for an internal standard. The testing sample, HHL, ferulic acid, captopril, and ACE were dissolved in ACE buffer (0.76 M boric acid, 0.2 M sodium hydroxide, and 0.3 M sodium chloride, pH 8.3). The area under the curve of hippuric acid (HA) was observed using UV–Vis RP-HPLC at a wavelength of 228 nm, 1 mL/min steady flow rate, and 15 min of isocratic elution (26% ACN with 0.1% TFA in deionized water). The inhibitory activity of ACE was determined using the following equation:(2)$$ACE\;inhibitory\;activity\;\left(\%\right)=\left[1-\left(\frac{{HA}_{\left(inhibitor\right)}}{{HA}_{\left(no\;inhibitor\right)}}\right)\right]\times 100\%,$$

The *HA_(inhibitor)_* and *HA_(no inhibitor)_* were the areas under the curve of the hippuric acid acquired in the presence and absence of an inhibitor. The half-maximal inhibitory concentration of ACE inhibition (ACEI IC_50_) was calculated using a non-linear interpolation constructed with a minimum of six logarithmic concentrations of the testing sample.

### 3.6. Determination of Dipeptidyl Peptidase-IV Inhibitory (DPP4I) Activity and Its Half-Maximal Inhibitory Concentration

The DPP4 inhibitory activity was determined as described by Nong, N. T. P. et al. [44], with a few modifications. The testing samples, GP-pNA, linagliptin, and DPP4 were dissolved in 0.1 M Tris-HCl buffer, pH 8.0. Briefly, 25 µL of the testing sample was premixed with 25 µL of 1.6 mM GP-pNA and incubated at 37 °C for 10 min. Then, 50 µL of DPP4 (0.5 U/µL) was added. The reaction was continued for 60 min at 37 °C. The enzymatic reaction was quenched by adding 100 µL of 1 M sodium acetate, pH 4.0. The absorbance of pNA, as a product of the DPP4 and GP-pNA reactions, was monitored using a PowerWave XS (Biotek^®^, Winooski, VT, USA) 96-well microplate reader at a wavelength of 405nm. DPP4 inhibitory activity was calculated by the following equation:(3)$$DPP4\;inhibitory\;activity\;\left(\%\right)=\left[1-\left(\frac{{pNA}_{\left(inhibitor\right)}}{{pNA\;}_{\left(no\;inhibitor\right)}}\right)\right]\times 100\%,$$

The *pNA_(inhibitor)_* and *pNA_(no inhibitor)_* were the absorbances of the pNA (λ = 405 nm) acquired in the presence and absence of an inhibitor, respectively. The half-maximal inhibitory concentration of DPP4 inhibition (DPP4I IC_50_) value was calculated using a non-linear interpolation for at least six logarithmic concentrations of the inhibitor.

### 3.7. Two Orthogonal Bioassay-Guided Fractionations of BGSP-GP

The peptides in the BGSP-GP hydrolysate were fractionated individually using reversed-phase high-performance liquid chromatography (RP-HPLC) and strong cation exchange (SCX) chromatography as described by Pujiastuti D. Y. et al. [45], with a few modifications. The RP-HPLC fractionation was performed using a UV–Vis HPLC (Chromaster™, Hitachi Co., Tokyo, Japan) system assembled with a Kinetex C18 column (Phenomenex Inc., Torrance, CA, USA, 4.6 mm × 250 mm, 5 µm). At a constant flow rate of 1 mL/min and a wavelength of 214 nm, the elution gradient was programmed as follows: a linear gradient of 14–35% B from 0–55 min and an isocratic elution of 35% B from 55–60 min. Solution A (5% ACN + 0.1% TFA) and solution B (95% ACN + 0.1% TFA) were used as the mobile phase. The RP-HPLC fractionation separated BGSP-GP hydrolysate into 12 fractions by collecting the eluate every 5 min. Meanwhile, the BGSP-GP hydrolysate was also fractionated using SCX chromatography. Self-packed SP Sepharose^®^ beads (Sigma-Aldrich, Saint Louis, MO, USA) were transferred into a micro-spin column (Pierce™, Thermo Fisher Scientific, Rockford, IL, USA) coupled with a peristaltic pump (LSP01-1A, LongerPump^®^, Heibei, China). The flow rate of SCX elution was programmed at 40 µL/min. The mobile phase comprised solution A (5% ACN and 0.2% FA) and solution B (5% ACN, 0.5 M sodium chloride, and 0.2% FA). The step elution of SCX fractionation was set as follows: 0% B (S1), 5% B (S2), 10% B (S3), 15% B (S4), 20% B (S5), 30% B (S6), 40% B (S7), 60% B (S8), 80% B (S9), and 100% B (S10). All the fractions were desalted using a HYPERSEP retain PEP C18 cartridge (Thermo Fisher Scientific, San Jose, CA, USA), freeze-dried, and the ACEI activity of each fraction was evaluated. Furthermore, the peptide sequences were identified using LC-MS/MS analysis from the most active RP-HPLC and SCX fractions.

### 3.8. Identification of Peptide Using LC-MS/MS Analysis Coupled with De Novo Peptide Sequencing and Database-Assisted Matching

Peptides in RP-HPLC fraction F5 and SCX fraction S1 were identified using a Q Exactive Plus Hybrid Quadrupole-Orbitrap mass spectrometer (Thermo Scientific Inc., Waltham, MA, USA) coupled with Dionex’s UltiMate 3000 RSLC nano system (Sunnyvale, CA, USA) and a nanoViper™ C_18_ (75 µm × 150 mm, 2 μm, 100 Å, PepMap™, Thermo Scientific Inc., Waltham, MA, USA) column. The liquid chromatography system was operated at a constant 0.25 mL/min flow rate. The elution gradient was programmed as follows: 0–5.5 min isocratic elution of 1% B, 5.5–45 min linear gradient elution from 1–30% B, 45–48 min linear gradient elution from 30–60% B, 48–50 min linear gradient elution from 60–80% B, and 50–60 min isocratic elution of 80% B, respectively. The mobile phase comprised solution A (0.1% FA in H_2_O) and solution B (0.1% FA in ACN). The high-resolution mass spectrometer (HRMS) was set in data-dependent acquisition (DDA) and positive ionization mode. Spray voltage 3.0 kV, sheath gas 40 arb, aux gas 5 arb, sweep gas 0 arb, capillary temperature 320 °C, s-lens RF 50, and heater temperature 300 °C were applied. The high-resolution mass spectrometer (HRMS) was operated in data-dependent acquisition (DDA) mode. The MS scan mass range window was opened at *m*/*z* 200–2000 with a resolution of 60,000. The MS2 fragmentation using 27% higher energy collision dissociation (HCD) was based on the 10th most abundant ions from the MS scan with a resolution of 30,000. The MS and MS2 spectra were analyzed using PEAKS X+ (Bioinformatics Solution Inc., Waterloo, ON, Canada). The *Momordica charantia* protein database from the NCBI protein database (https://www.ncbi.nlm.nih.gov, accessed on 7 May 2023) was imported. Peptide de novo sequencing and database matching were performed simultaneously in the PEAKS X+ software to acquire the peptide sequence. To add supporting confirmation of the identified peptide, the synthetic peptide, fraction F5 from HPLC, and fraction S1 from SCX were monitored for their retention time (t_R_) and MS spectra using an LCQ DECA XP MAX ion trap (Thermo Scientific Inc., USA) coupled with a Surveyor HPLC system (Thermo Scientific Inc., Waltham, MA, USA).

### 3.9. In Silico Analysis Prediction of Peptide Toxicity and Biological Activity

The biological potentialities of identified peptides were predicted using the PeptideRanker (http://distilldeep.ucd.ie/PeptideRanker/, accessed on 3 June 2023) [46] and BIOPEP-UWM (https://biochemia.uwm.edu.pl/biopep-uwm/, accessed on 3 June 2023) [47] databases. The toxicity prediction was carried out using the ToxinPred (http://crdd.osdd.net/raghava/toxinpred/, accessed on 3 June 2023) [48] database.

### 3.10. Synthetic APLVSW (AW6) Preparation and Purification

APLVSW (AW6) was synthesized in a peptide synthesizer by CEM Discover (CEM Co., Matthews, NC, USA) using Wang resin for the solid support, as described in our previous report [27]. The synthesis of AW6 was initiated from the C-terminal (Trp6) toward the N-terminal (Ala1) amino acid. The AW6 was released from Wang resin by adding TFA. Cold ether (−20 °C) precipitation and centrifugation at 10,000 rpm for 10 min were applied to extract AW6 from the peptide–TFA solution. Afterward, AW6 was purified using the same RP-HPLC system described in Section 3.7. At a constant flow rate of 1 mL/min under a wavelength of 214 nm, the elution gradient was programmed as follows: 0–5 min an isocratic gradient of 5% B, 5–20 min a linear gradient from 5–80% B, 20–25 min an isocratic gradient of 80% B, and 25–30 min an isocratic gradient of 5%. Solution A (5% ACN + 0.1% TFA) and solution B (95% ACN + 0.1% TFA) were used as the mobile phase. The peptide sequence was verified using an ion trap LC-MS/MS system.

### 3.11. Investigation of APLVSW (AW6) Inhibition Mechanism toward ACE and DPP4

The inhibition mechanism of AW6 towards ACE and DPP4 was investigated using the ACEI and DPP4I assays described above. The Lineweaver–Burk plot, also known as a double-reciprocal plot, was utilized to determine the Michaelis–Menten constant (K_m_) and maximum velocity (V_max_) as the *X*- and *Y*-axis intercepts of the primary plot, respectively. The Lineweaver–Burk plot consists of 1/substrate concentrations (1/S) on the *X*-axis versus 1/reaction velocity (1/V) on the *Y*-axis. The determination of the AW6 inhibition mechanism towards ACE was based on the production rates of hippuric acid (HA) in the absence and presence of AW6 (0, 10, and 15 µM) using five concentrations of HHL (0.3125, 0.625, 1.25, 2.5, and 5 mM) to generate the Lineweaver–Burk plot. The HA content in the reaction mixtures was quantified using the linear regression of the HA calibration curve. To construct the HA calibration curve, 0.08–2 mM HA solutions were used. At the same time, using five concentrations of GP-pNA (0.2, 0.4, 0.8, 1.6, and 3.2 mM) and three concentrations of AW6 (0, 0.15, and 0.2 mM), the pNA production rate was determined to evaluate the AW6 inhibition mechanism toward DPP4. Several pNA solutions (20–1000 µM) were prepared to generate the standard calibration curve of pNA. The pNA calibration curve was used to quantify the concentration of pNA in the testing mixtures.

### 3.12. Molecular Level Interaction Study of AW6 toward ACE and DPP4 through Simulation of Molecular Docking

Discovery Studio 3.0 software (Accelrys Software, UK) was utilized to rationalize the molecular-level interaction of AW6 toward ACE and DPP4. The crystal structures of human ACE–lisinopril (PDB Code = 1O86) and human DPP4–diprotin A (PDB Code = 1WCY) were obtained from the Protein Data Bank (https://www.rcsb.org/, accessed on 7 May 2023) and used as receptor targets. AW6, as a ligand, was prepared using Discovery Studio 3.0 software. Before the molecular docking simulation, all the ligands, water molecules, and inhibitors attached to the receptor except the enzyme cofactors were eliminated. The CHARMm force field (an integrated dynamic energy minimization and calculation program) was applied to the ligand and receptor. Blind docking simulation of AW6 toward ACE (pdb.1O86) was conducted with an SBD site sphere of 36.5 Å at the coordinates x: 39.26, y: 37.78, and z: 50.47. Meanwhile, the blind docking of AW6 toward DPP4 (pdb.1WCY) was carried out with an SBD site sphere of 32.7 Å at the coordinates x: 107.78, y: 57.5, and z: 42.12. The molecular docking simulation was executed in triplicate, and the best pose of receptor–AW6 was determined based on their CDOCKER energy score.

### 3.13. AW6 Stability toward Gastrointestinal Protease Digestion

The resistance of AW6 towards simulated gastrointestinal (SGI) digestion was evaluated using 1 mg/mL of AW6 in 35 mM sodium chloride buffer (pH 2.0), initiated by adding pepsin at an enzyme-to-peptide ratio of 2% (*w*/*w*). The hydrolysis reaction was conditioned at 37 °C for 1.5 h. Afterward, the pH was adjusted to 7.5 using 10 N NaOH. Then, α-chymotrypsin and trypsin were added at an enzyme-to-peptide ratio of 1:100 (*w*/*w*). The hydrolysis reaction was maintained for an additional three hours at 37 °C, and the stability of AW6 against SGI was evaluated using LC-MS analysis.

### 3.14. Multiple-Reaction Monitoring (MRM) Analysis of AW6 in Hydrolysate and Fractions

To quantify AW6 content in 1 mg of the testing samples, a TSQ Quantum Access Max (Thermo Scientific Inc., Waltham, MA, USA) is integrated with a heated-electrospray ionization (H-ESI) interface and coupled with a liquid chromatography system (Prominence UFLC, Shimadzu, Japan) equipped with a C_18_ column (Syncronis™, 150 × 2.1 mm, 5 µm, Thermo Scientific). The ionization was set to positive (+) mode with the following parameters: H-ESI temperature 300 °C, spray voltage 3500 V, sheath gas pressure 40 arb, aux gas pressure 5 arb, capillary temperature 320 °C, and tube lens offset 124 V. The MS scanning window was opened in a range of *m*/*z* 100–1000, and the MRM mode for the transition of *m*/*z* 672.4→282.2 with a collision-induced dissociation (CID) energy of 34 V was selected for the MS2 detection. The liquid chromatography separation at a 0.35 mL/min flow rate was carried out as follows: an isocratic elution of 5% solution B for 0–5 min; a linear gradient elution from 5% to 80% solution B for 5–20 min; an isocratic elution of 80% solution B for 20–25 min; and an isocratic elution of 5% solution B for 25–30 min, respectively. The mobile phase comprised solutions A (0.1% FA in H_2_O) and B (0.1% FA in ACN).

### 3.15. Statistical Analysis

The results were processed using GraphPad Prism 9.0 (GraphPad Software, Inc., La Jolla, CA, USA) and presented as the mean ± standard deviation from triplicate analyses. The data were analyzed using a one-way ANOVA with Duncan’s multiple range test (*p* < 0.05) post hoc procedure using SPSS 25.0 software (IBM, SPSS Inc., Chicago, IL, USA).

## 4. Conclusions

A potential ACEI peptide (APLVSW) with DPP4I activity was identified from BGSP-GP hydrolysate and screened using two orthogonal bioassay-guided fractionations coupled with in silico analysis. The ACEI and DPP4I IC_50_ values of APLVSW (AW6) were determined to be 9.6 ± 0.3 and 145.4 ± 4.4 µM, respectively. According to the molecular docking simulation and inhibition mechanism study, AW6 is a competitive ACEI and a non-competitive DPP4I peptide. The quantities of AW6 in 1 mg of BGSP-GP hydrolysate, fraction F5, and fraction S1 were 1.8 ± 0.2, 9.8 ± 0.3, and 2.5 ± 0.2 µg, respectively. Furthermore, AW6 is stable against SGI digestion, which indicates that AW6 might be resistant to the human gastrointestinal tract. This finding suggests that AW6 is promising as a potential ACEI peptide with DPP4I activity that could be beneficial for developing food supplements or therapeutic agents. However, further in vivo and clinical studies are indispensable for verifying the efficacy and bioavailability of this peptide.

## Figures and Tables

**Figure 1 pharmaceuticals-16-01629-f001:**
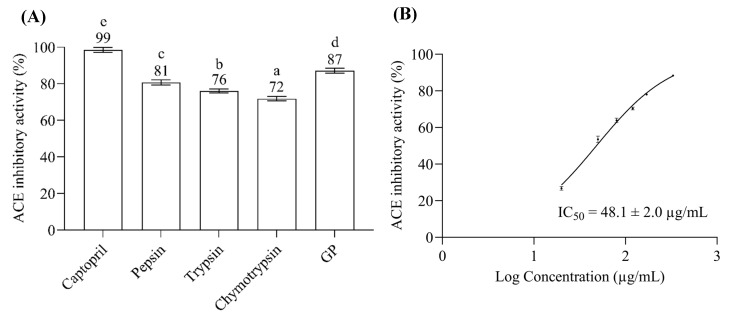
(**A**) ACE inhibitory activity of 1.6 µM of captopril as positive control and 0.33 µg/µL of BGSP hydrolysates generated by pepsin, trypsin, α-chymotrypsin, and gastrointestinal protease (GP) as testing samples. Different letters mean significant differences among testing samples (*p* < 0.05). (**B**) The half-maximal ACE inhibitory concentration of BGSP-GP hydrolysate.

**Figure 2 pharmaceuticals-16-01629-f002:**
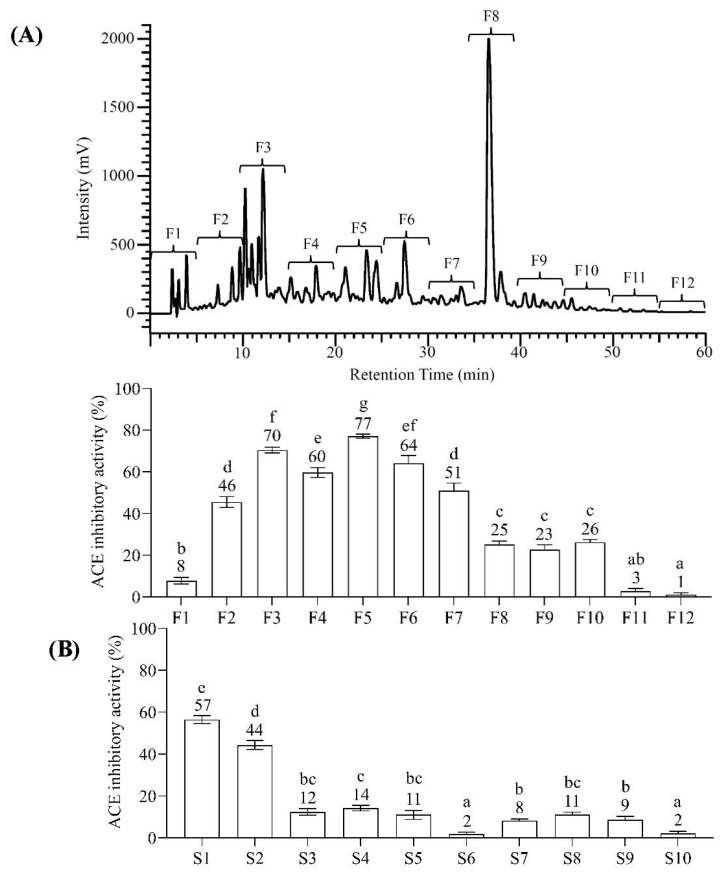
(**A**) Bioassay-guided RP-HPLC fractionation; (**B**) Bioassay-guided SCX fractionation. The concentration of each fraction was 0.17 mg/mL. Different letters mean significant differences among testing samples (*p* < 0.05).

**Figure 3 pharmaceuticals-16-01629-f003:**
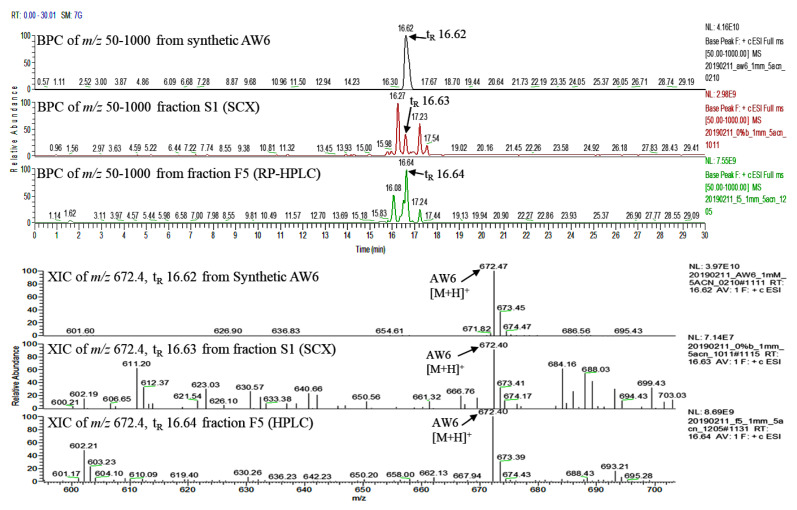
Base peak chromatogram (BPC) and extracted ion chromatogram (XIC) from LC–MS analysis of synthetic AW6 (**top**), fraction S1 from SCX (**middle**), and fraction F5 from RP–HPLC (**bottom**).

**Figure 4 pharmaceuticals-16-01629-f004:**
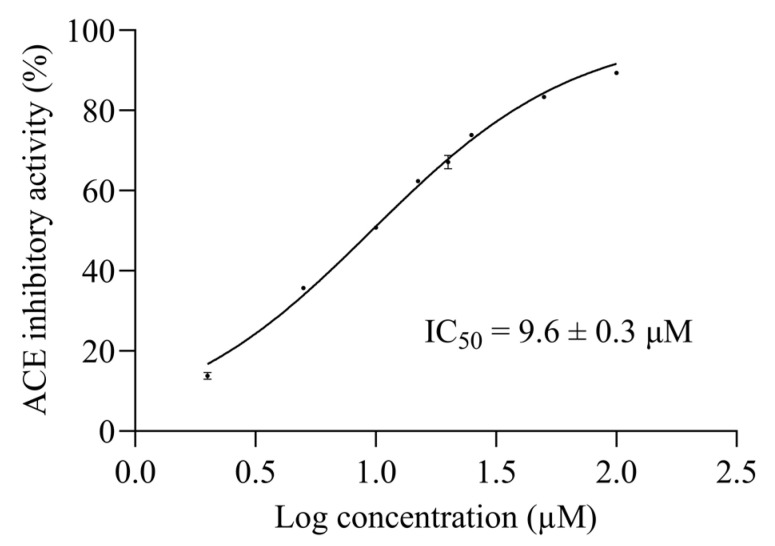
The half-maximal ACE Inhibitory concentration of AW6.

**Figure 5 pharmaceuticals-16-01629-f005:**
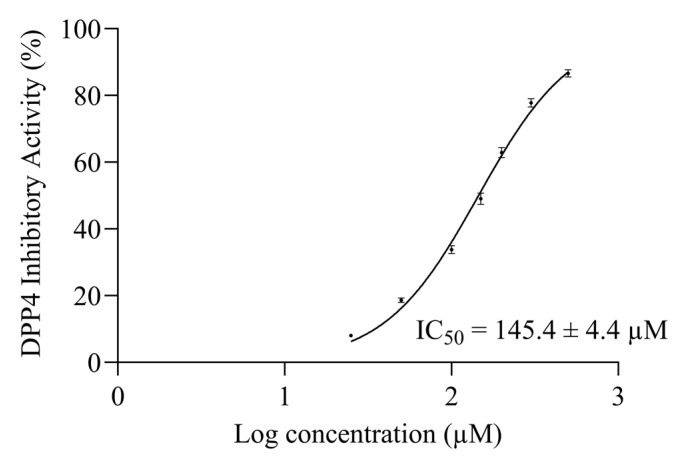
The half-maximal DPP4 Inhibitory concentration of AW6.

**Figure 6 pharmaceuticals-16-01629-f006:**
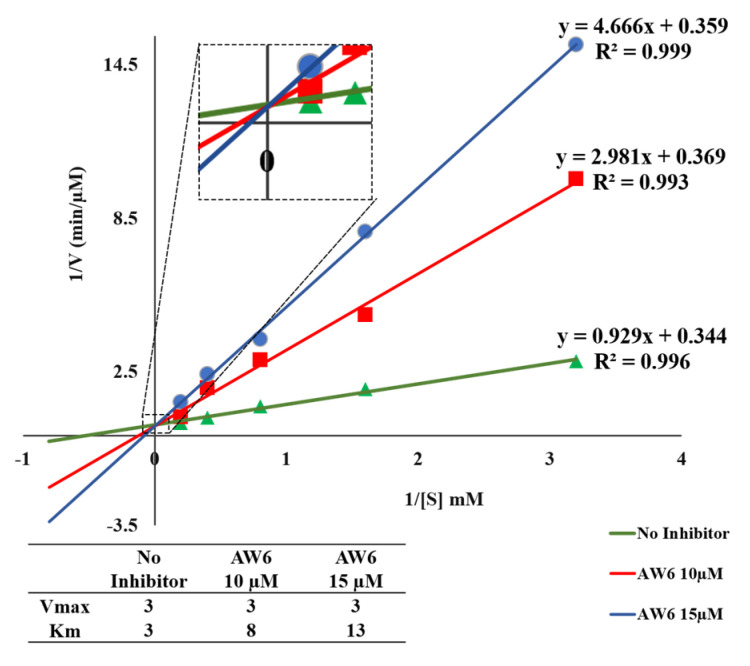
The double–reciprocal plot of AW6 toward ACE.

**Figure 7 pharmaceuticals-16-01629-f007:**
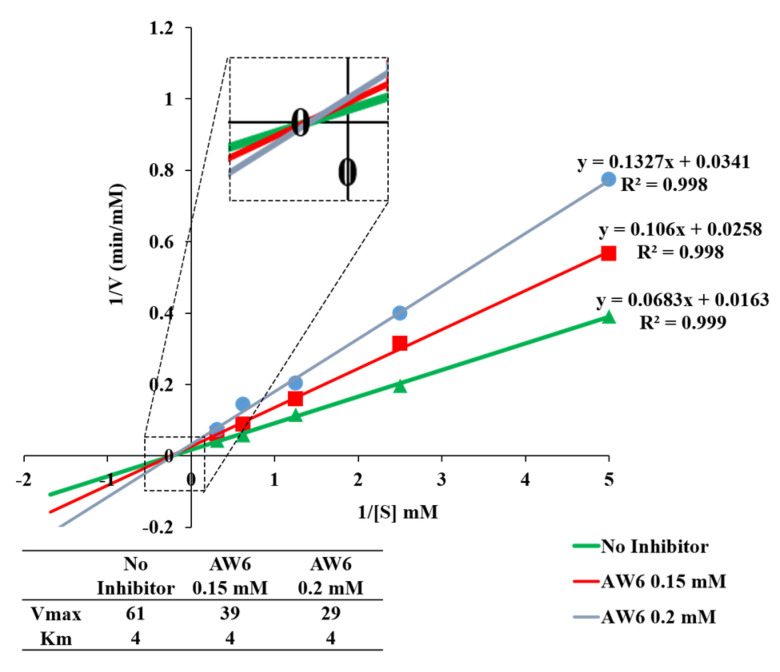
The double–reciprocal plot of AW6 toward DPP4.

**Figure 8 pharmaceuticals-16-01629-f008:**
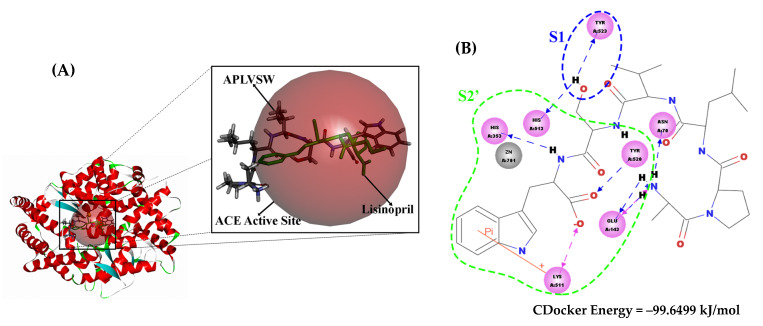
(**A**) Molecular docking study of AW6 and ACE (PDB code = 1O86) in 3D visualization. The red, transparent sphere represents the catalytic site of ACE. The green molecule is the lisinopril originating from the ACE–lisinopril complex (PDB code = 1O86), and the APLVSW (AW6) is visualized using a stick form with CPK atom color style. (**B**) The 2D visualization of intermolecular interaction between AW6 and ACE. The green and blue dashed lines indicate S2′ and S1 ACE active pockets that formed interactions with AW6, respectively.

**Figure 9 pharmaceuticals-16-01629-f009:**
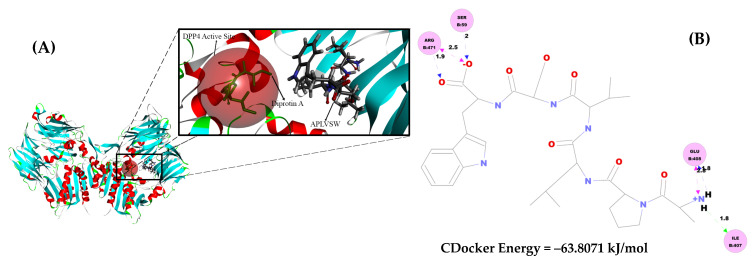
Molecular docking study of AW6 and DPP4 (PDB code = 1WCY). (**A**) The 3D visualization of how diprotin A interacts with the DPP4 catalytic site (the red, transparent sphere), and the APLVSW (AW6) forms interactions with DPP4 residues other than the catalytic sites. (**B**) The 2D visualization of intermolecular interaction between AW6 and DPP4.

**Figure 10 pharmaceuticals-16-01629-f010:**
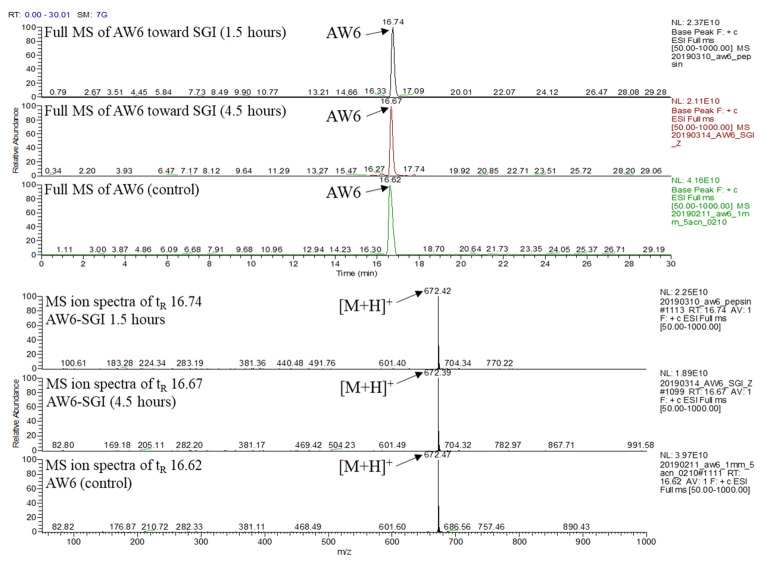
LC-MS chromatogram and the MS ion spectra of the stability of AW6 in simulation of gastrointestinal digestion.

**Table 1 pharmaceuticals-16-01629-t001:** In silico prediction of the ACEI peptide candidates from BGSP-GP hydrolysate fraction F5 (from RP-HPLC fractionation) and fraction S1 (from SCX fractionation).

Peptide Sequence	Peptide Length	ToxinPred	* BIOPEP (B)	PeptideRanker
APLVSW	6	Non-Toxin	0.00122	0.516023
EPTTSDVVVAGEFDQGSGSMR	21	Non-Toxin	0.00006	0.172646
ATISLENSW	9	Non-Toxin	-	0.191382

* BIOPEP (B): calculation of potential ACE inhibitory activity property of a protein (µM-1).

**Table 2 pharmaceuticals-16-01629-t002:** The amount of AW6 per milligram of testing sample.

	AW6 (µg)	Purification Fold
GP-BGSP hydrolysate	1.8 ± 0.2 ^a^	1.0
RP-HPLC fraction F5	9.8 ± 0.3 ^c^	5.4
SCX fraction S1	2.5 ± 0.2 ^b^	1.4

The different superscript letters represent a significant difference (*p* < 0.05) in AW6 quantity in the fractions compared to the GP-BGSP hydrolysate.

## Data Availability

Data is contained within the article and Appendix A.

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
