# Peer review of "Discovery and Characterization of a Dual-Function Peptide Derived from Bitter Gourd Seed Protein Using Two Orthogonal Bioassay-Guided Fractionations Coupled with In Silico Analysis"

_pharmaceuticals, 2023, doi:10.3390/ph16111629_

Round 1

Reviewer 1 Report

Comments and Suggestions for Authors

Manuscript “Discovery and characterization of a dual-function peptide derived from bitter gourd seed protein using two orthogonal bioassay-guided fractionations coupled with in silico analysis” is a good and informative study but had some minor corrections. Below are some comments/suggestion for the authors to improve its quality:

  1. Language should be improvised.
  2. Manuscript contains 30% similarity with already published article. Reduce the % of plagiarism upto 10%
  3. In Abstract section methodology of study should be included.
  4. In Abstract, authors should write at least two sentences about the materials and methods which have been used for this review article and sources that they did use to search for articles.
  5. Also abstract section should include some conclusive statements.
  6. Introduction section must include the need and significance of study.
  7. Materials and Methods should be rewrite.
  8. Conclusions: The conclusions are too general, format according to future aspects. Please make them more specific.
  9. Carefully read whole manuscript line by line and improve the sentence formation
  10. Please, add more updated references about the topic in different sections.
  11. Cross check all references and style of reference according to Journal format, use abbreviation of journal name in reference.

The work is interesting and worth publishing.

Comments on the Quality of English Language

Minor english correction required

Author Response

Dear Reviewer 1, 

We are deeply grateful for the comprehensive comments on our submitted manuscript, 'Discovery and characterization of a dual-function peptide derived from bitter gourd seed protein using two orthogonal bioassay-guided fractionations coupled with in silico analysis.' Your feedback is invaluable, and we sincerely appreciate your time and attention to reviewing our work. We have carefully addressed each comment in our response, provided in the attachment file. Please see the attached document for our detailed responses.

Thank you once again for your invaluable input, and we look forward to resubmitting the revised manuscript for your consideration.

Sincerely yours,

Dr. Jue-Liang Hsu 

Department of Biological Science and Technology
National Pingtung University of Science and Technology
1, Shuefu Road, Neipu, Pingtung 91201, Taiwan
E-mail: jlhsu@mail.npust.edu.tw
Tel: +886-8-7703202 Ext 5197; Fax: +886-8-7740550

Reviewer 2 Report

Comments and Suggestions for Authors

Authors have clearly planned and executed the work in a better way.

Molecular Docking part has to be revised, discussion about the active pockets, comparison with standard drugs is lacking.

Figure quality should be improved.

Avoid typo errors

Comments on the Quality of English Language

Its fine.

Author Response

Dear Reviewer 2, 

We are deeply grateful for the comprehensive comments on our submitted manuscript, 'Discovery and characterization of a dual-function peptide derived from bitter gourd seed protein using two orthogonal bioassay-guided fractionations coupled with in silico analysis.' Your feedback is invaluable, and we sincerely appreciate your time and attention to reviewing our work. We have carefully addressed each comment in our response, provided in the attachment file. Please see the attached document for our detailed responses.

Thank you once again for your invaluable input, and we look forward to resubmitting the revised manuscript for your consideration.

Sincerely yours,

Dr. Jue-Liang Hsu 

Department of Biological Science and Technology
National Pingtung University of Science and Technology
1, Shuefu Road, Neipu, Pingtung 91201, Taiwan
E-mail: jlhsu@mail.npust.edu.tw
Tel: +886-8-7703202 Ext 5197; Fax: +886-8-7740550

Reviewer 3 Report

Comments and Suggestions for Authors

The peptide was extracted from a hydrolysate that digested bitter gourd seed protein (BGSP) with digestive enzymes and consists of an amino acid sequence, APLVSW (AW6). In this study, the inhibitory potency and stability of AW6 against ACE and DPP4 were experimentally confirmed. The study is interesting, but there are some issues to amendment and consider.

1. Check whether the following experimental analysis method is correct. The IC50 of BGSP-GP hydrolysate is about 50 ug/ml, but it is awkward that UV-Vis RP-HPLC was performed at a high concentration of 50 ug/ul.

2. It is wondering as to how to distinguish between alphabetical order of statistical significance. It is suggested to mark a line between groups or use a general asterisk mark.

3. F3 and S2 are also considered as promising ACEIs. Discuss why these fractions were not compared with F5 and S1 fractions about DDP4 inhibitory activity.

4. There is information on the molecular weight of fraction (mean 110 g/mol) and AW6 (<3 kDa) being different. So, comparison data for the concentration of the fractions (S1 or F5) and the concentration of synthesized AW6 must be provided. If there is a significant difference, discuss the reason.

Comments on the Quality of English Language

Minor language polishing is required.

Author Response

Dear Reviewer 3, 

We are deeply grateful for the comprehensive comments on our submitted manuscript, 'Discovery and characterization of a dual-function peptide derived from bitter gourd seed protein using two orthogonal bioassay-guided fractionations coupled with in silico analysis.' Your feedback is invaluable, and we sincerely appreciate your time and attention to reviewing our work. We have carefully addressed each comment in our response, provided in the attachment file. Please see the attached document for our detailed responses.

Thank you once again for your invaluable input, and we look forward to resubmitting the revised manuscript for your consideration.

Sincerely yours,

Dr. Jue-Liang Hsu 

Department of Biological Science and Technology
National Pingtung University of Science and Technology
1, Shuefu Road, Neipu, Pingtung 91201, Taiwan
E-mail: jlhsu@mail.npust.edu.tw
Tel: +886-8-7703202 Ext 5197; Fax: +886-8-7740550

Reviewer 4 Report

Comments and Suggestions for Authors

The presented research concerns the characterization and biological tests combined with in silico analysis of the discovered bifunctional peptide derived from bitter gourd seed protein. The article is well written. My comments concern the introduction, which could be slightly modified and improved by providing information regarding the historical and ethnobotanical knowledge of the use and application of bitter gourd.

Moreover, it would be good if the authors indicated in the introduction the background of the discovered compound and its potential use in relation to current knowledge. You should highlight your discovery.

Also notes on the summary: These are mostly guesswork at this point. Please indicate specific conclusions regarding the research and findings conducted.

Moreover, the article is very interesting and arouses great interest. The research documentation and its presentation are very good.

Minor comments do not detract from the value of the presented research.

Author Response

Dear Reviewer 4, 

We are deeply grateful for the comprehensive comments on our submitted manuscript, 'Discovery and characterization of a dual-function peptide derived from bitter gourd seed protein using two orthogonal bioassay-guided fractionations coupled with in silico analysis.' Your feedback is invaluable, and we sincerely appreciate your time and attention to reviewing our work. We have carefully addressed each comment in our response, provided in the attachment file. Please see the attached document for our detailed responses.

Thank you once again for your invaluable input, and we look forward to resubmitting the revised manuscript for your consideration.

Sincerely yours,

Dr. Jue-Liang Hsu 

Department of Biological Science and Technology
National Pingtung University of Science and Technology
1, Shuefu Road, Neipu, Pingtung 91201, Taiwan
E-mail: jlhsu@mail.npust.edu.tw
Tel: +886-8-7703202 Ext 5197; Fax: +886-8-7740550

Reviewer 5 Report

Comments and Suggestions for Authors

In the manuscript, the authors document the identification and characterization of the peptide extracted from bitter gourd seeds. Using various analytical techniques, the authors demonstrate the bio-relevance (e.g., ACEI and DPP4I activities) of the identified peptide, and the reported data are of acceptable quality, but some figures need to be revised to improve readability. Below please find my comments for consideration. 

1. Revise the rounding to significant figures according to the ISO Guide to the Expression of Uncertainty of Measurement (GUM). It is awkward to write "145.4+/-4.39".

2. Revise lines 78-102. The text is more suited as part of the conclusion, but not the introduction. Suggest including text to signify the importance of the study.

3. Revise Figure 1B. Barely can see the error bars. By the way, what do the error bars represent? Standard deviation or relative standard deviation? Number of replicates, n=?

4. Revise Figure 3. The top panel should be labeled as "TIC of..." while the bottom should be labeled as "XIC of...". If the data were generated using the HRMS, individual m/z values in the bottom panel should have 4 decimal places. Otherwise what's point of using the HRMS. Make the two panels the same size.  

5. Figures 4&5. Barely can see the error bars. See comment#3.

6. Figures 6&7. These lines seem too linear to be true. Show individual data points on each line. Zoom in to show the intercept on the Y axis.  

7. Figure 8A and 9A. Part of the figure is truncated. 

8. Table 2. revise the rounding.

9. Section 3.8. What was used as the survey scan? Resolution settings are missing. Source temp? Voltage? Inclusion list?

10. Section 3.14, Why was this done using a different LC-MS? Why not use the LC-HRMS in Section 3.9?

11. Figure S4. Not informative, unless the authors included corresponding spectra to show fragmentation patterns associated with different collision energies. 

Comments on the Quality of English Language

Minor editing of English language required.

Author Response

Dear Reviewer 5, 

We are deeply grateful for the comprehensive comments on our submitted manuscript, 'Discovery and characterization of a dual-function peptide derived from bitter gourd seed protein using two orthogonal bioassay-guided fractionations coupled with in silico analysis.' Your feedback is invaluable, and we sincerely appreciate your time and attention to reviewing our work. We have carefully addressed each comment in our response, provided in the attachment file. Please see the attached document for our detailed responses.

Thank you once again for your invaluable input, and we look forward to resubmitting the revised manuscript for your consideration.

Sincerely yours,

Dr. Jue-Liang Hsu 

Department of Biological Science and Technology
National Pingtung University of Science and Technology
1, Shuefu Road, Neipu, Pingtung 91201, Taiwan
E-mail: jlhsu@mail.npust.edu.tw
Tel: +886-8-7703202 Ext 5197; Fax: +886-8-7740550

Round 2

Reviewer 3 Report

Comments and Suggestions for Authors

All issues have been well-addressed. There is no additional issue to raise.

Reviewer 5 Report

Comments and Suggestions for Authors

The authors have addressed reviewers' comments and revised the original version accordingly. Therefore, I would like to recommend the current version for publication.